# 'We don't see because we don't ask': Qualitative exploration of service users' and health professionals' views regarding a psychosocial intervention targeting pregnant women experiencing domestic and family violence

**Diksha Sapkota**[1,2,3]*, **Kathleen Baird**[1,4], **Amornrat Saito**[1,3], **Pappu Rijal**[5], **Rita Pokharel**[6], **Debra Anderson**[3,7]

**1** School of Nursing and Midwifery, Griffith University, Brisbane, Queensland, Australia, **2** Department of Nursing, Kathmandu University School of Medical Sciences, Dhulikhel, Kavre, Nepal, **3** Women's Wellness Research Program, Menzies Health Institute Queensland, Brisbane, Queensland, Australia, **4** Gold Coast University Hospital, Gold Coast, Queensland, Australia, **5** Department of Obstetrics and Gynaecology, B.P. Koirala Institute of Health Sciences, Dharan, Nepal, **6** College of Nursing, B.P. Koirala Institute of Health Sciences, Dharan, Nepal, **7** Faculty of Health, University of Technology Sydney, Sydney, Australia

* diksha.sapkota@griffithuni.edu.au, sapkotadiksha@gmail.com

**Data Availability Statement:** All relevant data are within the paper and its supporting information.

## Abstract

### Introduction

Given the relative recency of Domestic and Family Violence (DFV) management as a field of endeavour, it is not surprising that interventions for addressing DFV is still in its infancy in developing countries. In order to maximise the success of an intervention, it is important to know which aspects of the intervention are considered important and helpful by service providers and service users. This study, therefore, examined the acceptability of an antenatal-based psychosocial intervention targeting DFV in Nepal and explored suggestions for improving the program in future.

### Materials and methods

Intervention participants and health care providers (HCPs) were interviewed using semi-structured interviews. Data were audio-recorded and thematic analysis was used to analyse the data. Final codes and themes were identified using an iterative review process among the research team.

### Results

Themes emerging from the data were grouped into domains including perceptions towards DFV, impact of the intervention on women's lives and recommendations for improving the program. DFV was recognised as a significant problem requiring urgent attention for its prevention and control. Intervention participants expressed that they felt safe to share their feelings during the counselling session and got opportunity to learn new skills to cope with DFV. The majority of the participants recommended multiple counselling sessions and a

The original interviews cannot be shared because of ethical restrictions (issues of confidentiality of participants).

**Funding:** DS is supported by an International Postgraduate Research Scholarship (IPRS). The authors did not receive any additional funding for this work.

**Competing interests:** The authors have declared that no competing interests exist.

continued provision of the service ensuring the intervention's accessibility by a large number of women.

## Discussion

This is the first study to document the perspectives of women and HCPs regarding an ante-natal-based intervention targeting psychosocial consequences of DFV in Nepal. There was a clear consensus around the need to engage, support and empower victims of DFV and the intervention was well received by the participants. Ensuring good mental health and well-being among victims of DFV requires work across individual, organisational and community levels.

## Introduction

Domestic and Family Violence (DFV) has been identified as the greatest social epidemic of this modern era, with the problem at a higher magnitude in developing countries [1]. It is an increasing physical, psychological and economic health burden, affecting nearly 35% of women globally and 26% in Nepal [1, 2]. Being pregnant is a key risk factor for the beginning and/or escalation of DFV [2]. DFV during and around the time of pregnancy is associated with a wide array of mental health problems [3] and significantly poorer quality of life [4].

Nepal is among those countries, which have criminalised DFV as well as having adopted several policy and programmatic steps to better respond and prevent DFV [5, 6]. As of September 2018, One Stop Crisis Management Centres (OCMCs) have been established in 45 districts, with the aim of providing integrated services such as appropriate referrals and empowerment to survivors of violence [7]. However, recent studies have shown that only the most severe victims of DFV tend to visit these centres suggesting that most of these incidents go underreported [8]. Several bottlenecks both on the demand as well as supply sides were noted [5, 9]. For example, in terms of demand, because of the social stigma and fear associated with DFV, and lack of awareness, women are reluctant to seek support services (only 34% of women seek formal support services) [10]. Similarly, some of the supply or service side challenges such as lack of coordination between national bodies and local organisations as well as lack of sufficient resources make the programmes insufficient in number, irregular and typically short-term [5, 7]. Nonetheless, these government and non-government initiatives are certainly steps in a right direction, however, currently they are insufficient in number and have not been implemented properly [6, 7]. This protracted change provides a rationale to seek out an intervention that can be implemented at a low-cost and can be made accessible to a larger population of Nepal.

With a growing recognition of need and urgency to address DFV, several intervention strategies have been designed and implemented, particularly in developed settings [11, 12]. However, sufficient evidence does not exist to guide health care providers (HCPs) and policy makers on the best way to address the needs of victims. A recent review about interventions targeting DFV among pregnant women in low-and middle-income countries (LMICs) identified some beneficial effects of screening and supportive counselling with a provision of referral to support services [13], in alignment with previous findings [14, 15]. In a resource limited setting such as Nepal, where services for DFV are insufficient in number or are just emerging [8], screening for DFV alone can be a good initiative to inform women about DFV and the appropriate referrals of victims could contribute to an improvement in their health and well-being

[15, 16]. Antenatal care (ANC) has been the recommended setting for implementation of such a screening and referral intervention [13, 17].

Based on the available evidence and theories, a brief psychosocial intervention was designed to screen victims for DFV and address their immediate mental health needs. Furthermore, the intervention intended to support women in adopting safety behaviours and strengthening social support. The intervention was piloted in an ANC clinic within a tertiary hospital in Nepal. Pregnant women allocated to the intervention group were provided with a single session of counselling by a trained nurse, an information booklet and telephone support. The intervention nurse (DS), principal investigator of this study, is a trained counsellor and has past and present involvement in several research projects related to DFV. An intervention delivery guide was developed after several rigorous development and planning sessions within the research team and the intervention nurse adhered to this guide while delivering the intervention to the participants. The detailed description about the intervention and its delivery is published elsewhere [18]. Table 1 briefly illustrates the components of the intervention.

Employing both qualitative and quantitative research methods in trials has been greatly acknowledged in modern social research [19], particularly if the trial is related to complex health-related issues, such as DFV [20]. Besides developing a clinically effective intervention, it is also essential for interventions to be person-centred to maximise its effectiveness and engagement. Qualitative studies are a useful way of exploring participants' experiences of interventions and have often been used to understand the acceptability and engagement with those interventions [21]. Insights from qualitative research can inform if the implementation of an intervention is successful and can also identify any modifications desired by the participants to ensure its applicability and accessibility in the future [22]. Hence, a nested qualitative study was conducted with the following specific objectives:

**Table 1. Components of the piloted intervention and activities conducted.**

| Intervention component | Details of the activities conducted |
| --- | --- |
| DFV screening | Pregnant women (24–34 weeks of gestation) attending the ANC clinic were screened for presence of DFV including psychological, physical and sexual violence. |
| Supportive counselling | **IG:** Women were asked about their present concerns and were counselled accordingly. Brief information about DFV and its common mental health impacts were discussed. Women were provided with different problematic situations (based on the information booklet) and encouraged to express the way they would react in the given situation. Women's responses were supplemented with additional information based on the information booklet. Women were asked about the safety behaviours they were currently practising to prevent them from revictimisation, and the counsellor informed them about other possible alternatives to adopt in future. A single session of counselling was conducted by a nurse (DS) which lasted for 35–45 minutes. |
| | **CG**: Women were asked about their general health and wellbeing and were provided with key information on pregnancy and post-partum care. |
| Information Booklet | **IG:** Women were provided with the information booklet which included key information about DFV, approaches to address common mental health consequences of DFV, and safety planning behaviours. They were provided with the contact details of local support services against DFV. The booklet also included general information on pregnancy and postpartum care. |
| | **CG:** The information booklet included contact details of local referral services and general information on pregnancy and postpartum care. (Both booklets had same front cover and were given a neutral title to ensure confidentiality and participants' safety). |
| Telephone support | **IG:** Participants were provided with the contact details of the counsellor so that they could contact her at times of need at any time during the study period. |
| | **CG**: No contact information was provided. |

IG: Intervention group; CG: Control group

- To explore the perceived strengths and limitations of the counselling and psychoeducation intervention as experienced by intervention participants and HCPs.

- To solicit feedback on what elements of the intervention and process could be improved and how.

## Materials and methods

### Design

This study was nested within a trial conducted in an ANC clinic of a tertiary hospital in Nepal [23]. BP Koirala Institute of Health Sciences (BPKIHS) was selected purposively as it has wider catchment areas and nearly 100–150 pregnant women visit this hospital every day [23]. This trial has been registered in the Australian New Zealand Clinical Trial Registry (ANZCTR) with the registration number 12618000307202. A descriptive qualitative approach was implemented for this study. During initial assessment and participant recruitment, all participants in the intervention arm were asked if they would be willing to participate in qualitative interviews after receiving the intervention. All participants agreed to be interviewed. The study conduct and reporting adheres to the Consolidated Criteria for Reporting Qualitative Research (COREQ) [24]; see Supporting information S1 COREQ checklist.

### Data collection

Using an open-ended interview schedule, the research nurse (RP) asked women from the intervention group about the acceptability, strengths and weaknesses of the intervention during their follow-up visits, which occurred at: i) 4–6 weeks after the intervention; and ii) 6 weeks post-birth of a baby. DS (the intervention nurse) used a flexible interview topic guide for interviewing HCPs as it enabled a wider range of participants' views and experiences to be captured. In addition to the feedback regarding the piloted intervention, HCPs were asked about their views regarding DFV and efforts to address it in their context (see Supporting information S1 Interview schedule). Both interviewers (RP and DS) are registered nurses and have experience in conducting qualitative interviews. Purposive sampling was used to select the HCPs who could provide the most pertinent information related to the research topic and objectives. Interviews with the women lasted for about 15–20 minutes and with HCPs the length of each interview was on an average 30–45 minutes. Face-to-face interviews were conducted at the hospital; for women who were unable to come to the hospital, interviews were conducted via telephone. Adhering to the World Health Organization (WHO) ethical guidelines [25], the interviewer ensured that if the call was answered by the participant and it was safe for her to talk. If not, the woman was asked to call back at any time feasible for her. All the interviews were digitally recorded, fully transcribed verbatim and given a unique identification number to ensure anonymity of the study participants. Responses from the participants were also noted down on paper. Those notes were read in conjunction with the transcripts to ensure complete and appropriate interpretation of the information. All data were organised and managed using Microsoft Word and Excel spreadsheet.

### Data analysis

Data were analysed using inductive thematic analysis, consisting of six phases [26]. The phases were familiarisation with transcripts, forming initial codes, searching for themes, reviewing themes, defining and naming themes and developing report [26]. First, transcripts were read and reread to ensure familiarity. An initial coding frame was developed by two raters (DS and

RP) for three transcripts with the HCPs and 10 with the intervention participants. A coding framework was iteratively discussed among the researchers and any disagreement on codes were resolved on consensus. DS then coded remaining transcripts using the initial coding framework and discussed new codes with the other authors (DA, KB, AS) as they emerged, ensuring that the codebook was regularly updated. Coded chunks of data were initially grouped into categories and relevant themes, and all transcripts were read several times to capture all emerging themes and to extract supporting excerpts. Inconsistencies were discussed during joint meetings with the research team and themes were further developed [26]. The final stages of the analysis involved all authors reviewing and confirming the themes and subthemes in relation to the coded extracts, and ensuring that the analysis was logical and valid. Initial examination was made to assess for differences in the themes for interviews undertaken at first and second follow-up visits. However, the examination found no major differences by time-point and after discussion with the study authors, it was decided to present the data together as a whole piece.

## Trustworthiness/Rigour

The criteria described by Lincoln and Guba was applied to enhance trustworthiness in this study [27]. Credibility was enhanced by prolonged engagement, as the first author (DS) grew up, lived and worked in the research setting. By its very nature, qualitative research normally produces an enormous amount of data, requiring researcher interpretation, which was achieved by the researchers actively engaging with the text. A triangulation of researchers helped to provide diverse perspectives to the data. To strengthen trustworthiness and avoid missing any information during data analysis, hand notes were reviewed along with the transcripts. To stay closer to the text, the original Nepalese version was used in the coding process and translation into English took place once themes were established. The inclusion of the quotes in the findings allows readers to judge the interpretations made. All these attempts are believed to enhance the confirmability of the findings. A detailed description of the research processes used has ensured trustworthiness in the analysis and where appropriate, the findings can be transferred to other settings.

## Ethics approval and informed consent

The study received ethical approval from the Griffith University Human Research Ethics Committee, Nepal Health Research Council (NHRC), and Institutional Review Committee of BPKIHS. Written informed consent was obtained from all the study participants and they were assured that all their data would be treated confidentially and would be made non-identifiable. All participants were assured that they could withdraw from the study at any time. The study was conducted in adherence to the WHO's ethical and safety recommendations for research on violence [25].

## Results

### Characteristics of the sample

A total of 63 women at the first follow-up and 51 women at the second follow-up, belonging to the intervention group, were asked about their views and opinions regarding the intervention. Pregnant women (24–34 weeks of gestation) having a history of DFV were recruited in the study. A total of seven HCPs (3 nurses, 2 obstetricians/gynaecologist, 1 health manager, and 1 Midwifery Professor) were interviewed. Among the three nurses, one was a Prevention of Mother to Child Transmission (PMTCT) counsellor, one was a nurse in-charge of the

**Fig 1. Overview of domains, themes and subthemes.** This figure shows the three overarching domains and the related themes and subthemes that emerged from the analysis.

obstetrics and gynaecology outpatients department (OPD) and one was a nurse in-charge of the antenatal ward. A number of themes and subthemes were identified and they were grouped under several domains reflecting the research questions (Fig 1).

### Domain 1: Snapshot of Domestic and Family Violence (DFV)

HCPs were asked to comment on the burden and existing response mechanisms of DFV. This domain is divided into the following three themes.

**Highly prevalent, multifactorial and overlapping in nature.** HCPs acknowledged that DFV is a complex and significant issue in our setting. One respondent stressed that it is still very much considered as a hidden issue. Several of the HCPs commented that the women are often subjected to different forms of violence.

*"This is quite hidden issue in our society and women do not talk about this issue simply. Last year, we conducted research, where we found that more than 50% of women were victims of violence. Based on our study, we can assume that this is highly prevalent."* [HCP 01]

It was felt that the current societal system placed men above women, and this was considered as one of the important factors for DFV. Another respondent supported this and added a lack of education as another contributing factor for DFV.

*"A woman has to behave according to the direction of her father-in-law, her father, her umm. . . husband. . .and next is definitely illiteracy. . ."* [HCP 07]

*"Like I have already said, because of pressure from her mother-in-law (MIL) to have son, a woman needs to get pregnant again and again, her husband also thinks in the same way and there are cases when MIL has beaten her daughter-in-law because she does not work properly."* [HCP 06]

**Violence remains undisclosed, unasked and unshared.** In a male-dominated society of Nepal, DFV is often identified as a common family matter and as such, this violence although condoned can also be normalised. HCPs also believed that some forms of violence were common and probably had been experienced by all. Furthermore, they believed reporting every form of violence might lead to broken relationships. Some participants expressed these sentiments as follows,

*"It is not like that there is no violence. . .", "It is present in everyone's family. . .."* and *"Violence one or two times is persistent in majority of us".*

Another cultural norm is the belief that violence between a husband and wife is a private matter and should not be disclosed with others unless it turns out be a severe issue.

*"Taking those minor things directly to the police administration or calling them, asking for support in Maiti Nepal* [support organisation] *can break the relationship. I feel that is one of the important issues."* [HCP 03]

In the hospital where the research was conducted, there was no provision of routine antenatal enquiry for DFV. Therefore, without the routine provision of DFV enquiry by nurses or doctors, women accessing BPKIHS for their pregnancy care would not be offered an opportunity to reveal their history of violence.

*"At the time of ANC, while doing counselling we do not routinely ask about this problem. It has not yet become our part of care."* [HCP 01]

*". . .we have not seen such types of cases. It must be because we do not ask them. We do not have that facility and we do not ask them. There is no provision for asking anything related to violence . . ."* [HCP 06]

**Addressing DFV at different levels.** During interviews, HCPs did acknowledge Nepal government's current and ongoing commitment towards addressing DFV and securing a violence free life for women. At the time of the research, BPKIHS did not have OCMC, although the hospital management was lobbying with the government for its establishment. At the time the article was written, only rape victims presenting to the emergency department were referred to the police. The police were responsible for arranging for the medico-legal examination of victims and notifying the hospital's authorities. HCPs expressed concerns that after disclosure of abuse, the issues of social rehabilitation might arise. HCPs were concerned as there was a lack of support organisations available to provide ongoing support to the women, There was a general lack of awareness among the staff that for some women providing the space to talk and offering advice and support by HCPs may in itself be considered helpful.

*"In such cases* [separation from family], *where can we give security to her* [victim of violence]? *In which place we can keep her? We do not have such type of organisations. We just talk. . . ."* [HCP 05]

## Domain 2: Reflection of the program and its contents

Intervention participants and HCPs were asked to comment on how they perceived the different components of the intervention. Most of the participants expressed many benefits associated with the intervention, although some did express some barriers in adopting the intervention into their daily practice. The following two themes discuss these findings.

**Impact of intervention on women's lives.** Data from HCPs and women provided a nuanced and deeper understanding of some important and meaningful impacts of the intervention, which are summarised under the following two subthemes.

*A relationship based on trust.* Most of the women reported that the counsellor made them feel comfortable, displayed empathy and respect towards them, and spent time with them. They highlighted that this relationship of trust acted as a strong foundation in promoting their self-empowerment. Some women even verbalised that they felt they had actually found a friend in the counsellor. Women perceived the counselling session to be a safe and contained place where they felt safe to open up about their personal emotions, their concerns and experiences. The person-centred approach adopted during the counselling session allowed women to express their immediate concerns and motivated those unwilling to share and express openly.

*"I felt I had the confidence to come forward and express oneself. I felt relaxed and relieved."* [Woman, SR69]

*"Having discussions in a separate place in a confidential manner, I liked it. I liked getting advice and suggestions. We were taught according to our needs, so it was good."* [Woman, NR63]

*Feeling empowered, supported and valued.* Participants appreciated the proactive enquiry from the counsellor, which they considered promoted information sharing. They valued this approach as they had never been asked about such sensitive issues before.

*"I liked it as I was asked alone. Nobody has ever asked about such things. I always expect somebody to ask me such things. You asked, I felt really happy. I got one-to-one support."* [Woman, CL139]

*". . .we could say yes because we were asked. If we were not asked, we would not have known anything."* [Woman, NB136]

The majority of women were pleased to learn how to respond to DFV and found the advice offered by the counsellor as useful including "*safety behaviours*", "*support services*", and "*strengthening social support*". Though there was variation in the preferences for the particular topic areas among participants, they all valued the individualised approach adopted by the counsellor. Participants expressed that the intervention helped them by promoting their capacity for self-enhancement and adopt strategies to keep themselves safe and healthy in future.

*"I can share* [the information related to conflict with in-laws] *openly with my husband. I felt happy, developed courage. I was unable to speak initially. Now, my relationship with my husband has improved.* [My] *self-efficacy has enhanced."* [Woman, SD99]

*"There was important information such as keeping documents and money safely. I have now kept my and my children's copy of certificates at my parents' house as well."* [Woman, NP66]

HCPs further highlighted that providing education to women can be a cost-effective strategy as they can act as change agents by transmitting their learning to others, especially to other female family members and friends.

*"They can share the information they have known with other daughters, daughters-in-law or their relatives or someone in need in their village or in neighbourhood. If needed, they can give the numbers."* [HCP 05]

The majority of participants expressed feeling reassured with a sense of continued support when they were provided with the contact details of the counsellor.

*"Even if they cannot come on their own, by providing the contact numbers, they can decrease their emotional feeling to some extent."* [HCP 05]

Most of the participants verbalised that the booklet was easy to use, and the illustrations helped them to understand the content. Some of the common phrases used by the participants to explain the booklet was "*easily understandable*", "*simple language*", "*comprehensive*", "*good illustrations*" and "*illustrations complemented the text*". The feature of the booklet that women seemed to appreciate the most was the promotion of self-learning, and the opportunity to share it with others who are interested in learning about DFV.

*"I understood a number of things after reading the book on my own."* [Woman, BS74]

*"There were lots of information that I did not know earlier. I read that booklet repeatedly. Knowing simple tips regarding how to get rid of stress has really helped me a lot."*[Woman, CL139]

*"Phone numbers included in the booklet can be shared with others as well. Once, I gave the number of police station to one of my friends who said that her husband beat her when he was drunk."* [Woman, NR63]

**Perceived barriers to the intervention implementation.**   Despite the positivity towards the feasibility of the intervention occurring within a healthcare setting, some participants expressed some practicability issues related to the intervention. For instance, HCPs were circumspect about the provision of contact details of support services as this might lead to over reporting of the incidents of DFV which could result in escalation of broken family relationships. Similarly, four out of seven HCPs were worried about the potential safety related threats associated with the disclosure of DFV. Although, the DFV enquiry was always made in a private place, some women were concerned that they might face an interrogation from their family members about what were they asked in the clinic by the HCPs. Hence, family awareness of discussing violence with others could have negative repercussions.

*"Family members might force her to tell why she had been taken inside, what she was asked. Oh, you have gone* [disclose information to HCP] *no? Did your organisation do anything? What did that organisation do? Telling these things, the woman might be re-victimised and one thing is there will be discrimination."* [HCP 06]

One participant was concerned that disclosing everything to HCPs might cause problem in future.

*"I have told you everything trusting you, but sometimes get scared that it might cause some bad impacts in future."* [Woman, DK100]

Whilst most women acknowledged the booklet as a valuable learning resource, some participants felt the booklet was difficult to read because of the volume of content, and the small font and pictures contained within it. Another barrier highlighted by some women was 'time'; they felt they were just too busy with their day-to-day activities to use the booklet regularly.

*"Though the book includes information that was previously unknown, we do not have much leisure time to read the book."* [Woman, LR39]

Some barriers when using the phone as expressed by participants were "*poor network*", "*lost the number*", "*not always feasible*" and "*hard to talk about sensitive issues over the phone*". Indeed, this was also raised by one HCP who specifically mentioned that contacting women via telephone or expecting them to make contact by telephone should not be the preferred method, as it did not allow for the face-to-face therapeutic communication between the counsellor and the woman, and can lead to feelings of insecurity.

*"Rather than having telephone conversation, if we can meet her personally for interview, then it would be more effective, or we can talk a little more, we can guess from her looks as well."* [HCP 05]

*"It is not always easy and feasible to discuss about family related matters over the phone."* [Woman, MC135]

Furthermore, during pregnancy, participants' commented that their immediate needs were related to pregnancy and childbirth and they wanted to focus on the future and try to put their experiences of DFV behind them.

*"It's not easy to leave him. I am scared that my child's future will be threatened."* [Woman, DK100]

### Domain 3: Recommendations for improving the intervention in the future

As the discussion turned to how best to implement the intervention in future, women and HCPs offered some ideas for its improvement and continuation. It was evident that the participants appreciated the comprehensive nature of the intervention; however, they presented some clear ideas on how the intervention could be improved, which are presented in the following four themes.

**Expanding the reach of the program.** Participants suggested extending the program into other health settings to address the issue of accessibility of the current program as only those women visiting BPKIHS had access to the program. It was thought that advertising the service widely through the hospital would be an excellent strategy to increase the uptake of the service by a larger cohort of people.

*"As this study was done in a single setting, many women who do not visit this particular health facility are not included in the study. We are aware to some extent, that is why we are here, but what about others who are not allowed to seek health service?"* [Woman, YK01]

Offering a continued delivery of the service in the health facility was desired by most of the participants. Indeed, some women also made recommendations about including non-pregnant women in the intervention. Few women (n = 3) suggested distributing the booklet to a wider network could help those who are unable to or do not want to come to health centres.

**Embedding the intervention into routine health care.** The majority of the HCPs were in favour of the integration of the DFV package into routine health care or PMTCT counselling service. They felt this as a cost effective way to benefit a large proportion of women in dealing with DFV. However, a nurse in-charge was not supportive of the idea of integration; she believed victims of DFV have varied needs and expectations and therefore, they require specialised support, which should be dealt separately from routine care.

*"The program is different in itself, otherwise integration can also be possible. At first, listening too much information at a time they* [women] *might not feel* [interested]. *They may not feel like saying about these things, so in that case I feel it would be good to have different program."* [HCP 06]

**Committed staff, sufficient resources and supportive management.** HCPs recognised the importance of offering support to women experiencing DFV, and appreciated the intervention and expressed a willingness to continue with the program. However, they firmly asserted that for its successful implementation, they required an extra consultation room as well as trained personnel.

*"We need a separate counsellor, a separate room to deliver the intervention. It will be difficult to deliver in the existing infrastructure."* [HCP 04]

HCPs stressed that initiating a program is not enough, what is important is a commitment to the continuation of the program. The success of the program is dependent upon knowledgeable and committed staff and continual support from senior management.

*"Initiating a program is not just enough. There should be integration, after conducting such program there is minimal follow-up to assure that the actions are being implemented as planned."* [HCP 05]

HCPs highlighted several factors for improving the motivation among staff and the success of the program, which include appropriate workload and protected time for initial and continuation of training. All HCPs felt that collaboration with stakeholders at administrative, local, district and national level was also vital to the implementation of DFV program. Regular supportive supervision to staff were suggested by some HCPs to allow for discussing gaps in knowledge and developing strategies to implement the program efficiently.

*"To start immediately, we need a willingness among staff. Starting from HOD* [Head of Department]*, all higher authorities need to be involved. If this can be done, there is no thing that it* [the intervention] *cannot be done."* [HCP 05]

**Adoption of different modalities of care.** DFV is a complex issue and addressing it requires multi-dimensional and multi-sectoral health interventions. Participants suggested several recommendations for modifying the program and adopting new strategies, which are grouped under the following two subthemes.

*Changing the structure of the program.* Both HCPs and women indicated that there should be multiple and ongoing support sessions as they believed having a regular interaction facilitates disclosure and greater retention of the content taught.

*". . .spending some time together or had there been multiple visits, then more issues might have been unfolded, because in one single visit, not everything will come out."* [HCP 04]

*"There can be some people who are reluctant to disclose at first and will disclose later only. If meeting can be made time and again, there will be easiness in sharing own feeling."* [Woman, KR61]

Some of the participants recommended group teaching and sharing and role-play, as they believed such approaches promote disclosure and encourage a deeper and active engagement from women.

*"Role-plays in a group would be helpful to make us understand the matter easily."* [Woman, AB36]

*". . .involving victims of violence in a group and providing counselling. For example, when I said what I had felt, others may also say that these things have also happened to me."* [HCP 06]

One clinician suggested adopting a shared care model [General Practitioner (GP) or/and Midwifery care model] could be a viable strategy in distributing the workload, and such a model might help in ensuring the best quality of care to those seeking care.

*"It is not necessary that every pregnant woman is examined by a gynaecologist or obstetrician. Even in abroad, midwives look after them or they are examined by GP and only high-risk cases are referred to us. If this can be done, we will have decreased patient load and they* [pregnant women] *will also receive adequate time."* [HCP 04]

*Holistic approach in addressing DFV.* HCPs as well as women believed that awareness raising at an individual level is not sufficient to deal DFV effectively. They suggested family/husband involvement for better and effective dealing with the issue.

*"These days everyone knows about violence, but just knowing the violence does not mean that change lies on one's hand."* [Woman, CC94]

*"I do not know how this problem can be solved. It is not possible to change family members on our own. Some of the elderly are uneducated and it is difficult to make them understand."* [Woman, NR63]

*"At the same time, we need to provide education to the family. It is essential to inform those perpetrators that doing such things is wrong."* [HCP 01]

A small proportion of women felt that contextual factors, such as household and parenting responsibilities, family interference and expectations limited their attendance and adherence to the intervention. As a potential solution to this, they proposed home-based counselling service.

*"Please continue this program. It would be better if villages are visited and help is provided to the victims of violence. For those who do not come to hospital and stay at home, even providing them with this booklet would be helpful."* [Woman, SR96]

A peer education model was also proposed, believing that peer involvement would ensure that the women would receive help and support from a person they know and at a place that they are familiar with.

*"Teaching someone and asking her to teach others as well can help many to learn. Giving one or two extra books and asking that person to provide the booklet to other one or two people that she knows."* [Woman, NB136]

*"Providing awareness in the community that the information should be shared openly would make many to understand the problem. If the program is conducted by including some active women from the community, it would be more effective."* [Woman, BR130]

From the interviews, it was evident that women understood the complexity of the problem and therefore believed multiple strategies were required to respond to the problem and meet the individual needs of the women and their families.

## Discussion

Qualitative exploration of participants' feelings added valuable insights to the design and effectiveness of a health sector based psychosocial intervention trialled in a resource-constrained setting. This study revealed both the strengths and weaknesses of several components of the intervention and identified the most useful and relevant outcomes for the participants. Results of this study can be used as a reference for developing and implementing a nurse-led intervention in addressing DFV in LMICs.

HCPs perceived DFV as a significant problem and reflected on the social drivers, such as patriarchal norms and illiteracy, behind such violence. Similar findings were reported in other studies from Nepal [2, 9]. Participants believed the intervention to be an empowering, easy and beneficial process. A supportive and therapeutic relationship between the women and HCPs facilitated disclosure and the exploration of violence; this was supported by findings from other studies [28, 29]. This uninhibited expression of oneself led to an increase in confidence, self-esteem and knowledge among women. Women further expressed a better ability to adopt safety strategies, seek social support and cope with mental consequences of DFV, similar to another study [22]. While participants acknowledged the use of an individualised approach in relation to a woman's circumstances and varying informational needs [30] and reported its beneficial effects, the wider literature showed the short-term effects of such counselling intervention [14, 31]. Hence, studies with longer follow-up are recommended to conclude about the sustained effects of the counselling-based intervention. Some participants reported practicability issues in relation to regular use of the booklet and accessing and making the telephone call, as they felt they have other day-to-day family commitments to deal with rather than DFV. Therefore, it is advised to consider alternative method of information sharing in further studies.

For complex issues such as DFV, a one-size-fits-all approach usually does not work, as different people have different needs [32]. As a multitude of factors beyond the individual level influence DFV, participants expressed a need of adopting a holistic approach reaching across the relational, community and organisational levels of the social ecology [33]. Participants in this study proposed some strategies such as home-based counselling, husband involvement and peer involvement for improving response against DFV. Though home-based counselling and peer involvement have shown beneficial impacts in supporting victims and helping them to utilise support services [34], there were differing perspectives on the involvement of

husband during counselling [35]. Particularly, in a patriarchal society such as Nepal, involving husband in the counselling might actually escalate the risk of future violence and isolate the woman even further. Therefore, it is suggested that involving husbands should be considered as an adjunct to individual therapy and couple counselling should be directed towards discussions around gendered roles, healthy relationships and responsible parenthood rather than directly discussing about DFV [35]. In addition, such approaches have been tested in developed countries, and thus, may not truly reflect the developing context. Hence, it is advisable to continually develop and evaluate context-specific interventions for addressing DFV in LMICs.

In the current study, participants also suggested delivering the intervention through multiple sessions; however, current literature indicates that the greatest improvements occur during the initial days of the intervention and decrease over time [36]. Considering the time and resource constraints, and evidence from past studies demonstrating the significant effects of single session therapy (SST) in improving the mental health and safety behaviours of abused women [37, 38], SST is considered a pragmatic and the best option. In addition, interventions requiring multiple interactions with women was pointed out as an important reason for high drop-out rates [39] and low attendance rates were seen in subsequent sessions [14, 40].

Screening in the antenatal period would provide an opportunity in creating awareness regarding potential causes and effects of DFV, which is crucial to protect the survivors and hold perpetrators accountable. Although screening at an ANC clinic has been significantly linked to increased identification of victims of DFV [40, 41], screening alone does not necessarily help victims to access services [16]. In the excerpts provided in the first domain in this study, it was evident that HCPs considered themselves unprepared to challenge the existing norms supporting violence and take action to help the victims of DFV. However, appropriate and regular training will help HCPs to develop communication skills that will help them to empower women to disclose their experiences of DFV openly and take appropriate actions to provide support and manage the consequences of DFV [8, 30]. Training of HCPs should include issues such as understanding dynamics of a violent relationship and socio-cultural drivers of DFV, validating women's experiences, screening and dealing with barriers faced during screening, maintaining information regarding community resources and assisting victims with appropriate referrals [42, 43].

In agreement with previous studies, competing demands, time constraints and lack of sufficient resources were mentioned as the hindering factors for delivering DFV interventions in healthcare settings [28, 29]. System level changes such as training of HCPs, staff commitment, institutional support, onsite support services and effective collaboration with local stakeholders involved in providing support services are important for delivering continual and effective service to victims of DFV [40, 43, 44]. Considering the social stigma related to seeking help from DFV services and minimal funding in this area [8, 10], integrating DFV program into routine ANC and/or PMTCT counselling services can be a potential cost-effective strategy to improve response mechanisms against DFV in resource-constrained settings like Nepal. Such integrated approach of service delivery has been recommended in literature as well [13, 41].

## Methodological considerations

The main strength of this study is seeking feedback from all participants, which enabled a broader capture of participants' views and ensured general representation of the overall sample. Throughout the process, measures to ensure trustworthiness, essential for others to judge the value of the study, have been taken [27]. Despite being a novel study contributing to bridge the knowledge gaps, it must be acknowledged that this study has some limitations. Due to the safety concerns, it was not possible to capture the reasons for discontinuation of the study by

the participants (7 women at the first follow-up and 19 at the second follow-up were lost to follow-up). However, these drop-out rates are comparable with similar research [14, 40] and the intervention appeared to be effective in addressing important needs of the victims. Furthermore, as this study included only women accessing one health facility for their antenatal check-up, it did not represent the overall expectations and attitudes of women visiting other health facilities. As a consequence, the findings of the study may be transferrable to similar groups but they cannot be generalised.

## Conclusion

This study identified elements of the intervention that participants regarded as either beneficial or problematic, thereby providing insights on how the intervention may be improved for future use. Disclosure of violence and seeking support services was facilitated by a relationship of trust between service users and HCPs. The intervention was perceived as an innovative and effective approach in addressing the immediate needs of victims of DFV, however, for effective dealing with DFV, participants suggested expanding the intervention to a wider setting and delivering it through multiple sessions. Integration of the intervention within routine ANC package was proposed as a potential strategy to improve the accessibility and sustainability of the intervention. Overall, this study emphasised the need for education and organisational support to create a supportive environment to facilitate engagement and knowledge among victims of DFV.

## Supporting information

**S1 COREQ checklist. This supporting file includes the COREQ checklist of the present study.**
(DOCX)

**S1 Interview schedule. This supporting file includes the interview schedule for intervention participants and health care providers.**
(DOCX)

## Acknowledgments

The authors would like to thank BPKIHS team for providing us an opportunity to conduct this study in this hospital. Furthermore, we are very grateful to all HCPs and women for their active participation in the interviews. Our sincere thanks to Nicole McDonald and Stephanie Zietek for reviewing and editing the English language and grammar of the final version of the manuscript.

## Author Contributions

**Conceptualization:** Diksha Sapkota, Kathleen Baird, Debra Anderson.

**Data curation:** Diksha Sapkota, Amornrat Saito, Rita Pokharel.

**Formal analysis:** Diksha Sapkota, Kathleen Baird, Rita Pokharel.

**Investigation:** Diksha Sapkota, Rita Pokharel.

**Project administration:** Diksha Sapkota.

**Resources:** Pappu Rijal, Debra Anderson.

**Supervision:** Kathleen Baird, Amornrat Saito, Pappu Rijal, Debra Anderson.

**Validation:** Kathleen Baird, Debra Anderson.

**Writing – original draft:** Diksha Sapkota.

**Writing – review & editing:** Kathleen Baird, Amornrat Saito, Pappu Rijal, Rita Pokharel, Debra Anderson.

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
