## [Decision Letter · Decision Letter 0]

18 Oct 2019

PONE-D-19-22252

‘We don’t see because we don’t ask’: qualitative exploration of service users’ and health professionals’ views regarding a psycho-social intervention targeting pregnant women experiencing domestic and family violence

PLOS ONE

Dear Mrs. Sapkota,

Thank you for submitting your manuscript to PLOS ONE. After careful consideration, we feel that it has merit but does not fully meet PLOS ONE’s publication criteria as it currently stands. Therefore, we invite you to submit a revised version of the manuscript that addresses the points raised during the review process.

We would appreciate receiving your revised manuscript by Dec 02 2019 11:59PM. To enhance the reproducibility of your results, we recommend that if applicable you deposit your laboratory protocols in protocols.io, where a protocol can be assigned its own identifier (DOI) such that it can be cited independently in the future. For instructions see: http://journals.plos.org/plosone/s/submission-guidelines#loc-laboratory-protocols

We look forward to receiving your revised manuscript.

Kind regards,

Susan Bartels, MD, MPH, FRCPC

Academic Editor

PLOS ONE

Additional Editor Comments:

Thank you for your important work on violence against women in Nepal. The article has been independently critiqued by three reviewers who have provided feedback below. In particular, the authors are asked to provide a more critical appraoch to the intervention (feasibility, barriers, efficacy, etc.) and to clarify some of your methodologies.

2.

We suggest you thoroughly copyedit your manuscript for language usage, spelling, and grammar. If you do not know anyone who can help you do this, you may wish to consider employing a professional scientific editing service.  

3. Please ensure you have included the registration number for the clinical trial referenced in the manuscript.

4. We note that you have included your manuscript title page as a separate, 'Other' file.

Please ensure that you include a title page within your main document. You should list all authors and all affiliations as per our author instructions and clearly indicate the corresponding author.

Reviewers' comments:

Reviewer's Responses to Questions

**Comments to the Author**

1. Is the manuscript technically sound, and do the data support the conclusions?

Reviewer #1: Yes

Reviewer #2: Yes

Reviewer #3: Partly

2. Has the statistical analysis been performed appropriately and rigorously? 

Reviewer #1: N/A

Reviewer #2: N/A

Reviewer #3: N/A

3. Have the authors made all data underlying the findings in their manuscript fully available?

Reviewer #1: No

Reviewer #2: Yes

Reviewer #3: No

4. Is the manuscript presented in an intelligible fashion and written in standard English?

Reviewer #1: Yes

Reviewer #2: Yes

Reviewer #3: Yes

5. Review Comments to the Author

Reviewer #1: Overall, the study is an important contribution to the literature and to practice of domestic violence prevention. A few revisions are recommended below for consideration to enhance the study’s influence.

1. The abstract mentions that the study will assess feasibility and efficacy. However, both need additional articulation as to what realms of both of these issues the qualitative study was meant to assess and how do the findings relate to those domains. For example, feasibilty could be assessed in a much broader realm than what is presented. The same for the efficacy results. What were the primary outcomes and how did the study reach or not reach those goals as ascertained through the qualitative assessment. While this is an embedded qualitative study, there were most likely primary outcomes for feasibility and efficicacy that would be helpful to state up front and assess the accomplishment of the study against those expectations.

2. Needed, especially in the discussion but also in the introduction is a more critical assessment of the role of screening in settings where services are perceived to be lacking or ineffective. There is growing programming in Nepal from the government and from local and international NGOs that deal with this topic, but still the services were percevied to be to lacking. More contextualization of what is available versus what is perceived to be available and effective would help the reader to ascertain this disconnect.

3. Similarly, given the perceived lack of services, and possibility of actual lack of services, it would be helpful to have a more indepth discussion of screening and referral in such an environment. This has programmatic implications but also ethical ones that would be helpful to understand in more detail.

4. The discussion has a strong bend toward literature that supports the study’s findings. A more critical assessment of the literature would useful to help situation the study’s findings in the broader programmatic and research literatures.

5. Additionally, a stronger critique of a one-session intervention is needed. The respondents for the most part seemed to suggest that more is better and that is often the case with exposure to interventions, including literature supporting interventions for health sector based violence prevention. While the intervention is described as theory based, it would be helpful to understand what kind of change was anticipated from a one session counseling session, that the respondents seemed to indicate was useful, but not sufficient to change practices that are both normative and perpetrated by others.

6. The health providers mentioned the importance of the topic, but also clearly indicated the fear of marriage dissolution. This speaks to the strong norms underpinning the family as a key unit as well as the fear of massive over-reporting, which is not likely the case given statistics the world over that a vast minority of violence survivors seek formal help, especially through the health sector. Additional consideration of these perceptions would be helfpul to understand what type of training and support is needed at the facilitator level to make screening and referral successful.

7. The health providers mentioned several institutional barries to making this type of programming work. Additional consideration as to the feasibility of this type of intervention with limited staffing, limited time, etc in a lower income setting. While there is a move to have one stop centers in hospitals, how widespread is this and how likely is continued funding for the staffing and infrastructure needed to sustain the program as sustainability is a key theme of the analysis.

8. The short duration of the interviews with survivors and the lack of quotes that showed how survivors actually used the brochures or the counseling in their real life suggests a lack of depth of the impact. Is there evidence that represents more than general perceptions of utility and benefit? This would be helpful evidence to see impact/efficacy beyong more general perceptions of utility.

9. The study particpants represent individuals who in large part participated in the program. A significant number did not and there is no information about those individuals. In terms of feasibility and efficacy, what does the non participant/lack of follow up suggest?

Reviewer #2: Thank you for the opportunity to review this manuscript - it addresses a very important topic. The research design for the study was appropriate and there is evidence of a robustly undertaken study. The study was well contextualised in the context of DFV in LIMC and the activities of the intervention was useful. I have the following observations for the authors to improve the manuscript:

Abstract

In the abstract (conclusion) we are informed for the first time that the intervention was an antenatal intervention. this information would have been useful to have earlier in the abstract. The 'introduction' would be an appropriate place.

It would be useful for terminology to be consistent throughout. The term 'abuse' is used interchangeably with 'violence'.

Data Collection

Data were collected from the women at 4-6 weeks after the intervention and gain 6 weeks post birth. This indicates that there was some kind of longitudinal data collection. On lines 96-97 it is noted that “verbal and non-erbal cues from the participants were noted down on paper as well”. This would not be for all participants as some were interviewed by telephone. Clarity needed. How were these observations used as they are not discussed in the analysis section?

Data collection at 2 time points with the women allows for some investigation of whether or not their views changed over time, and more nuanced understanding of how they used the information provided across time. The analysis, however, makes no distinction between data that were collected shortly after the intervention and post-birth. It would also be interesting to know whether the same questions were asked at each time point or if they changed and the reason for either. For a longitudinal data collection, I would have expected to see an analysis with a temporal component.

Data Analysis

The reference that is provided for the data analysis is Braun and Clarke, yet the description of the process that was undertaken does not reflect the 6 steps that Braun and Clarke identify. Additionally, the description of data analysis does not appear to reflect steps that were taken in the order they were undertaken. It is also noted that “the final codebook was developed through progressive iterations with other authors” (lines 103-104). This reads as if the emerging data did not impact on the codebook. Taken to its logical conclusion, this could suggest that data were ‘moulded’ to fit the codebook and not vice versa. I’m certain that this was not what the authors intended. Greater clarity around data analysis is needed.

Trustworthiness/Rigour

The reference provided for Lincoln and Guba is Cohen, Crabtree, yet the weblink does not appear to have Cohen & Crabtree on the site (Lincoln and Guba are, but I could not see Cohen & Crabtree). Clarification needed. Additionally Lincoln and Guba use the term 'confirmability' not 'conformability' - clarification needed.

Results

Domain 2 has a very nice lead in sentence about who was asked about the issues to be discussed. In Domain 1, understanding that it was just HCPs who were asked about DFV per se is less clear. A sentence similar to Domain 2 would be very useful.

Discussion

On page 25 consideration is given to including husbands in the counselling sessions with the women. I would urge caution here given the large literature about the dangers to women in doing this, or at the very least, include the suggestion but with the caveat around the literature on the potential dangers and inappropriateness of such an approach in the literature.

General comments

It would be useful for more details about the counsellor(s) to be included in the manuscript. It is not until line 471 that the reader is told that the intervention is 'nurse-led'. I have taken this to mean that the 'counsellor' is a nurse? This needs explanation and incorporation earlier in the manuscript. It would also be important to include the level of understanding of the counsellor of the dynamics of violent relationships, particularly bearing in mind the body of literature that describes the inappropriate response that many women who are experiencing DFV receive from counsellors who do not have this understanding.

The quotes often appear a bit disconnected from the text that introduces and describes them. Consider interspersing your quotes rather than presenting them in a block at the end of each section.

On page 12 (lines 220-221) it is noted that ……[the approach] ………helped them [women] to consider adopting a change in their lives”. I’m sure it is just the way it is written, but it reads as if the onus if on the women to change rather than the perpetrator to end his use of violence.

Page 16 (among others) mention is made of making contact by telephone (line 298). The authors do note that the WHO ethical and safety recommendations for violence related research were utilised, but it might be worthwhile making a specific comment about how you were able to be assured that the woman was safe, or her safety would not be compromised, if you were making contact by telephone.

Page 21 – lines 411 – 413 provides a brief description of s shared care model. Please consider including within that text that the clinicians need to understand the dynamics of a violent relationship – having clinical knowledge is insufficient.

Page 21 (lines 423-424) reference is made to an intervention that addresses perpetrators. Whilst this may be a worthwhile intervention it is a very different intervention to what was piloted. This needs to be recognised.

Typographical errors and errata

Page 4 – delete ‘the’ between ‘about’ and ‘DFV’

Page 4 - change ‘practicing’ to ‘practising’

Page 5 – delete comma after ‘Besides’

Page 9 – add ‘s’ to ‘explanation’

Page 12 – change’ spend’ to ‘spent’

Page 18 – the term ‘participant’ is used as a descriptor for a quote. Change?

Page 20 – it is noted (line 391) that addressing DFV required ‘multi-dimensional health interventions’ Not only do interventions need to be multi-dimensional, they need to be multi-sectoral. Include?

Page 25 – change ‘setting’ to ‘settings’ (line 513)

Awkward expression:

Lines 143 – 144: “Data analysis materialised a number of themes and subthemes”

Line 494 – change “punish the perpetrators” to “hold perpetrators to account”

Lines 505-507 – re-word “Similar to the current finding……….support services”

Reviewer #3: This paper is in an area of great importance because there is a dearth of interventions for IPV especially in low income countries. Process evaluations of trials of interventions are vital and not done enough in research. There are several areas for improvement in the manuscript detailed below.

Abstract

It is not clear what the research questions are or specific objectives. The themes are a bit difficult to interpret

Introduction is a good background and the objectives are clear here.

Methods. It is not clear when the second interviews were done in relation to the intervention as women were recruited up to 34 weeks?

how many people in total were in the intervention group- was it 63?

what was the pool of HCPs that the 7 were selected from and how were they able to provide the most pertinent information?

What were the interview questions?

Were the women and HCPs analysed together? and why?

Please spell out the Braun and Clarke method in more detail.

Results

It is confusing the domains and then the themes and they are not reflected in the abstract clearly. Further the materail needs to be synthesised more there are too many themes. A further synthesis would strengthen the analysis.

The themes are very descriptive, which does not match quality thematic analyses. It appears that the domains might reflect direct interview questions which has resulted in this very descriptive level analyses?

Why is the first domain only from HCPs?

Please remove some of the aconyms

In the second domain it would be good to get a sense of the strength of the themes- some women, most women....

p12 it is not helpful to discuss the quantitative findings as they are not presented here.

Some of the quotes are very powerful but sometimes the headings for the subthemes don't match the quotes e.g. new and positive learning experience starts with the value of being asked alone and the feeling empowered section suddently has barriers using the phone?

Discussion

The discussion repeats the findings quite a lot rather than summarising and the conclusion is not helpful as we dont have the quantitatve results.

6. PLOS authors have the option to publish the peer review history of their article (what does this mean?). If published, this will include your full peer review and any attached files.

Reviewer #1: No

Reviewer #2: Yes: Prof Colleen Fisher

Reviewer #3: No

---

## [Author Response · Author response to Decision Letter 0]

21 Nov 2019

Response to reviewers’ comments

I would like to extend my sincere thanks to the editor and the reviewers for careful and thorough reading of this manuscript and for their thoughtful comments and constructive suggestions. I have revised the manuscript in the light of the provided feedback and comments and highlighted text indicates the changes made. Responses to their specific comments/suggestions/queries are as follows:

[note: C: Comment, R: Response]

S.N. Editor Comments to Author: Responses

C1. Please ensure that your manuscript meets PLOS ONE's style requirements, including those for file naming. The PLOS ONE style templates can be found at

R1 Thank you for your informative feedback. The necessary amendment has been made.

C2. We suggest you thoroughly copyedit your manuscript for language usage, spelling, and grammar. If you do not know anyone who can help you do this, you may wish to consider employing a professional scientific editing service. 

R2. The research team consisted of person who are native English speakers and the manuscript has been thoroughly reviewed and revised by them. In addition, language usage, spelling and grammar were thoroughly edited by two colleagues, who are native English speakers and have sound knowledge in research.

1. Nicole McDonald, Project Manager, My Health for Life

2. Stephanie Zietek, Senior Research Officer, Women’s Wellness Research Program

All the requested information/files have been uploaded.

C3. Please ensure you have included the registration number for the clinical trial referenced in the manuscript. 

R3: Thank you. The registration number has been included in the manuscript. (Page no. 7, line no: 125-126)

C4. We note that you have included your manuscript title page as a separate, 'Other' file.

Please ensure that you include a title page within your main document. You should list all authors and all affiliations as per our author instructions and clearly indicate the corresponding author. 

R4.Thank you. Title page has been included in a main document and it aligns with the manuscript guidelines (Page 1)

 Responses to reviewers’ comments 

A. Reviewer #1: Overall, the study is an important contribution to the literature and to practice of domestic violence prevention. A few revisions are recommended below for consideration to enhance the study’s influence.

C1. The abstract mentions that the study will assess feasibility and efficacy. However, both need additional articulation as to what realms of both of these issues the qualitative study was meant to assess and how do the findings relate to those domains. For example, feasibility could be assessed in a much broader realm than what is presented. The same for the efficacy results. What were the primary outcomes and how did the study reach or not reach those goals as ascertained through the qualitative assessment? While this is an embedded qualitative study, there were most likely primary outcomes for feasibility and efficacy that would be helpful to state up front and assess the accomplishment of the study against those expectations. R1. Thank you for your comment. This was a nested qualitative study and the main aim of this study was to explore the experiences of participants regarding the intervention and their perceived impacts. Objectives have now been written in a simpler and clearer terms in the abstract section. (Page 2, line no 25-28)

C2. Needed, especially in the discussion but also in the introduction is a more critical assessment of the role of screening in settings where services are perceived to be lacking or ineffective. There is growing programming in Nepal from the government and from local and international NGOs that deal with this topic, but still the services were perceived to be to lacking. More contextualization of what is available versus what is perceived to be available and effective would help the reader to ascertain this disconnect. 

R2. Thank you for your constructive feedback. A brief account of the ongoing programmes and efforts for addressing DFV in Nepal has been included in the introduction section. In addition, gaps or limitations of the current activities targeting DFV in Nepal are also highlighted. (Page 4: line no 59-75)

In the introduction and discussion section, role of screening in resource-constrained setting has been written. (Page 5, line no. 83-86 & Page 26, line no 544-548)

C3. Similarly, given the perceived lack of services, and possibility of actual lack of services, it would be helpful to have a more in-depth discussion of screening and referral in such an environment. This has programmatic implications but also ethical ones that would be helpful to understand in more detail. 

R3.Thank you for your constructive feedback. A brief account of the ongoing programmes and efforts for addressing DFV in Nepal has been included in the introduction section of the paper. In addition, gaps or limitations of the current activities targeting DFV in Nepal are also highlighted. (Page 4: line no 59-75)

In the last 2 paragraphs of the discussion section, the measures to improve screening and referral in the study setting has been discussed and supported by literature. (Page no. 26-27, line no 544-568)

C4. The discussion has a strong bend toward literature that supports the study’s findings. A more critical assessment of the literature would be useful to help situation the study’s findings in the broader programmatic and research literatures. 

R4. Thank you for your constructive feedback. Additional literature was reviewed and statements supporting the programmatic implications of the findings have now been added in the discussion section (Page 26-27, line no. 544-568)

C5. Additionally, a stronger critique of a one-session intervention is needed. The respondents for the most part seemed to suggest that more is better and that is often the case with exposure to interventions, including literature supporting interventions for health sector based violence prevention. While the intervention is described as theory based, it would be helpful to understand what kind of change was anticipated from a one session counseling session, that the respondents seemed to indicate was useful, but not sufficient to change practices that are both normative and perpetrated by others. 

R5. Thank you for your feedback. Given the time and financial constraints, one session intervention was the only viable option and several articles have supported the use and effectiveness of single session therapy in addressing violence. 

These points have been included in the discussion section (Page no: 26, line no. 536-543) 

Furthermore, the objective of this intervention was to help victim understand that DFV is a problem and help is available against it and also help her develop strategies to cope with the negative consequences of violence. This intervention did not intend to change the gender and societal values and behaviours of perpetrators, for which, we definitely will need an intervention delivered in multiple sessions and over the long run. 

C6. The health providers mentioned the importance of the topic, but also clearly indicated the fear of marriage dissolution. This speaks to the strong norms underpinning the family as a key unit as well as the fear of massive over-reporting, which is not likely the case given statistics the world over that a vast minority of violence survivors seek formal help, especially through the health sector. Additional consideration of these perceptions would be helpful to understand what type of training and support is needed at the facilitator level to make screening and referral successful. 

R6. Thank you for your feedback. 

The main things to be included while training HCPs to make screening and referral successful are now mentioned in the discussion section (Page no. 26-27, line no 551-557).

C7. The health providers mentioned several institutional barriers to making this type of programming work. Additional consideration as to the feasibility of this type of intervention with limited staffing, limited time, etc in a lower income setting. While there is a move to have one stop centers in hospitals, how widespread is this and how likely is continued funding for the staffing and infrastructure needed to sustain the program as sustainability is a key theme of the analysis. 

R7. The situation analysis of availability and performance of OCMCs is included in the introduction section (Page 4, line no 60-73). 

Recommendations from HCPs and women regarding approaches in making the program sustainable have been discussed in the discussion section (Page no: 26-27, line no: 544-568).

C8. The short duration of the interviews with survivors and the lack of quotes that showed how survivors actually used the brochures or the counselling in their real life suggests a lack of depth of the impact. Is there evidence that represents more than general perceptions of utility and benefit? This would be helpful evidence to see impact/efficacy beyond more general perceptions of utility. 

R8. Thank you for your feedback. We do acknowledge the short duration of interviews with the participants. However, as the women were asked to provide their feedback about the intervention after they finished their quantitative follow-up assessments, it was not feasible to have long in-depth interviews with them. Furthermore, we only intended to seek their feedback regarding strengths and weakness of the intervention only. 

Quotes reflecting the impact/ benefits of the intervention as perceived by the participants are added under the theme “Impact of intervention on women’s lives”. (Page no 15-16, line no. 308-314) & (Page no: 17, line no: 333-338)

C9. The study participants represent individuals who in large part participated in the program. A significant number did not and there is no information about those individuals. In terms of feasibility and efficacy, what does the non participant/lack of follow up suggest? 

R9. Although there was a 10% of participant drop-out in first assessment and about 27% in second follow-up assessment, this was comparable with similar research. This information has been added to the discussion [Page no. 27, line no. 575-577]. It does indicate that there are some for whom the program was not feasible or acceptable. However, due to the safety of the women, we were unable to capture reasons for their loss to follow-up. As such this is listed as a limitation of the study in lines 575-579 on page 27-28. However, the program does still appear to be effective for the women who were able to be captured and as such addresses important needs for them. 

B. Reviewer #2: Thank you for the opportunity to review this manuscript - it addresses a very important topic. The research design for the study was appropriate and there is evidence of a robustly undertaken study. The study was well contextualised in the context of DFV in LIMC and the activities of the intervention was useful. I have the following observations for the authors to improve the manuscript:

C1. Abstract

In the abstract (conclusion) we are informed for the first time that the intervention was an antenatal intervention. this information would have been useful to have earlier in the abstract. The 'introduction' would be an appropriate place.

It would be useful for terminology to be consistent throughout. The term 'abuse' is used interchangeably with 'violence'. 

R1. Thank you for the feedback. 

To make it clear, the abstract has been rewritten and sentence explaining that the intervention was an antenatal intervention was added.

(Page 2, line no: 25-28)

Manuscript was read carefully, and the terminology ‘violence’ has been used throughout the manuscript to ensure consistency.

C2. Data Collection

Data were collected from the women at 4-6 weeks after the intervention and again 6 weeks post birth. This indicates that there was some kind of longitudinal data collection. On lines 96-97 it is noted that “verbal and non-verbal cues from the participants were noted down on paper as well”. This would not be for all participants as some were interviewed by telephone. Clarity needed. How were these observations used as they are not discussed in the analysis section?

Data collection at 2 time points with the women allows for some investigation of whether or not their views changed over time, and more nuanced understanding of how they used the information provided across time. The analysis, however, makes no distinction between data that were collected shortly after the intervention and post-birth. It would also be interesting to know whether the same questions were asked at each time point or if they changed and the reason for either. For a longitudinal data collection, I would have expected to see an analysis with a temporal component. Non-verbal cues were only collected from women who were interviewed in person. While these were initially examined, they did not contribute any additional information. As such, since this information was not included we have removed this line to help reduce confusion. (Page 8, line no: 150-151)

R2. The primary purpose of this study was not to observe changes in views over time. However, as you mentioned the temporal change could have been an interesting question to examine. As the same questions were asked at each time-point, an initial examination was made for differences in the themes for each time-point. This examination found no major differences by time-point and after discussion by the author team it was decided to present the data together as a whole piece. A sentence regarding this has been added to the data analysis section (Page 9, line no: 169-172).

C3. Data Analysis

The reference that is provided for the data analysis is Braun and Clarke, yet the description of the process that was undertaken does not reflect the 6 steps that Braun and Clarke identify. Additionally, the description of data analysis does not appear to reflect steps that were taken in the order they were undertaken. It is also noted that “the final codebook was developed through progressive iterations with other authors” (lines 103-104). This reads as if the emerging data did not impact on the codebook. Taken to its logical conclusion, this could suggest that data were ‘moulded’ to fit the codebook and not vice versa. I’m certain that this was not what the authors intended. Greater clarity around data analysis is needed. 

R3. Thank you. Six phases of thematic analysis as suggested by Braun and Clarke is now mentioned in the analysis section. The codes were discussed iteratively with the research team. Codes were grouped into categories and potential themes were extracted. After several rounds of discussions with the research team, the themes were finalised and supporting excerpts extracted.

The data analysis section has been rewritten with further detail providing a deeper explanation of 6 steps of Braun and Clarke (Page no. 9, line no.: 154-172).

C4. Trustworthiness/Rigour

The reference provided for Lincoln and Guba is Cohen, Crabtree, yet the weblink does not appear to have Cohen & Crabtree on the site (Lincoln and Guba are, but I could not see Cohen & Crabtree). Clarification needed. Additionally, Lincoln and Guba use the term 'confirmability' not 'conformability' - clarification needed. 

R4. Thank you for your comment.

Initially we have used the secondary source given by Cohen and Crabtree in their webpage. However, following feedback, reference has been changed to the original book Lincoln and Guba (Reference no. 26).

Sorry for the typographical error. The correct terminology ‘confirmability’ has now been written (Page 10, line no. 184).

C5. Results

Domain 2 has a very nice lead in sentence about who was asked about the issues to be discussed. In Domain 1, understanding that it was just HCPs who were asked about DFV per se is less clear. A sentence similar to Domain 2 would be very useful. 

R5. Intervention participants were asked to provide answer to open-ended questions regarding strengths and weakness of the intervention, and any recommendation they might have during their follow-up assessments. At this point, it was not appropriate to ask about the participants’ views regarding DFV and its response mechanisms. 

However, in-depth interviews were conducted with the HCPs and question regarding how they perceive the problem of DFV and the existing efforts to address were asked at first as it was deemed necessary to explore their views regarding DFV before asking them about their views regarding the piloted intervention. 

Hence, Domain 1 included views expressed by HCPs only. A sentence has been added in the Methods: Data collection section (Page 8, line no 138-139) and a next sentence has been added in Results: domain 1 to make it clear (Page 11, line 211).

C6. Discussion

On page 25 consideration is given to including husbands in the counselling sessions with the women. I would urge caution here given the large literature about the dangers to women in doing this, or at the very least, include the suggestion but with the caveat around the literature on the potential dangers and inappropriateness of such an approach in the literature. R6.Thank you for your feedback. 

This has been mentioned in the discussion section and supported by relevant literature (Page 25, line no 525-531). 

C7. General comments

It would be useful for more details about the counsellor(s) to be included in the manuscript. It is not until line 471 that the reader is told that the intervention is 'nurse-led'. I have taken this to mean that the 'counsellor' is a nurse? This needs explanation and incorporation earlier in the manuscript. It would also be important to include the level of understanding of the counsellor of the dynamics of violent relationships, particularly bearing in mind the body of literature that describes the inappropriate response that many women who are experiencing DFV receive from counsellors who do not have this understanding. 

R7. Thank you for your feedback.

The intervention was delivered by a trained nurse who has several years of experiences in working with victims of violence. Furthermore, the nurse had adhered to the intervention delivery guide to ensure consistency while delivering the intervention to all participants allocated to the intervention. This has been explained briefly in the introduction section (Page 5-6, line no. 93-99)

The detailed description about the research team including the intervention nurse is included in the protocol paper (Sapkota D, Baird K, Saito A, Rijal P, Pokharel R, Anderson D. Counselling-based psychosocial intervention to improve the mental health of abused pregnant women: a protocol for randomised controlled feasibility trial in a tertiary hospital in eastern Nepal. BMJ Open. 2019;9(4):e027436.)

C8. The quotes often appear a bit disconnected from the text that introduces and describes them. Consider interspersing your quotes rather than presenting them in a block at the end of each section. 

R8. Thank you for your suggestion. We have tried to intersperse the quotes where feasible and appropriate (Page no 14-24). In some instances where findings are summarised in one or two sentences, in order to preserve the essence of the message we want to deliver, we have included the quotes at the end of that particular section only.

C9. On page 12 (lines 220-221) it is noted that ……[the approach] ………helped them [women] to consider adopting a change in their lives”. I’m sure it is just the way it is written, but it reads as if the onus if on the women to change rather than the perpetrator to end his use of violence. 

R9. Thank you. The sentence has been restructured to make it clearer.

C10. Page 16 (among others) mention is made of making contact by telephone (line 298). The authors do note that the WHO ethical and safety recommendations for violence related research were utilised, but it might be worthwhile making a specific comment about how you were able to be assured that the woman was safe, or her safety would not be compromised, if you were making contact by telephone. 

R10. Thank you. The statement focusing on strategy adopted to ensure safety to participant while making telephone call has been added in the methods section under sub heading Data Collection (Page 8, line no 145-148). 

In the original manuscript, it was mentioned that the study was conducted in adherence to WHO’s ethical guidelines (Page 10, line no. 184-186). The approaches adopted to ensure safety and confidentiality to participants based on WHO ethical and safety recommendations for conducting intervention research on violence against women have been explained in detail in protocol paper. (Sapkota D, Baird K, Saito A, Rijal P, Pokharel R, Anderson D. Counselling-based psychosocial intervention to improve the mental health of abused pregnant women: a protocol for randomised controlled feasibility trial in a tertiary hospital in eastern Nepal. BMJ Open. 2019;9(4):e027436.)

C11. Page 21 – lines 411 – 413 provides a brief description of s shared care model. Please consider including within that text that the clinicians need to understand the dynamics of a violent relationship – having clinical knowledge is insufficient. 

R11. Thank you for your comment. 

In the discussion section, key issues that needs to be included while training HCPs on dealing with violence have been mentioned (Page 26-27, line no. 553-557).

C12. Page 21 (lines 423-424) reference is made to an intervention that addresses perpetrators. Whilst this may be a worthwhile intervention it is a very different intervention to what was piloted. This needs to be recognised. 

R12. Thank you for your constructive feedback. This was based on participants’ feedback, where they recommended of having program for perpetrators as well. Based on this single comment only, we can’t recommend conducting DFV program for perpetrators. 

Potential consequences of involving husband in counselling section has been discussed in brief in discussion section (Page 25, line no 525-531).

C13. Typographical errors and errata

Page 4 – delete ‘the’ between ‘about’ and ‘DFV’

Page 4 - change ‘practicing’ to ‘practising’

Page 5 – delete comma after ‘Besides’

Page 9 – add ‘s’ to ‘explanation’

Page 12 – change’ spend’ to ‘spent’

Page 18 – the term ‘participant’ is used as a descriptor for a quote. Change?

Page 20 – it is noted (line 391) that addressing DFV required ‘multi-dimensional health interventions’ Not only do interventions need to be multi-dimensional, they need to be multi-sectoral. Include?

Page 25 – change ‘setting’ to ‘settings’ (line 513) 

R13. Thank you for pointing out these errors. 

All mentioned typographical errors were addressed. In addition, the manuscript has been critically reviewed by authors and language and grammatical errors are now corrected.

C14. Awkward expression:

Lines 143 – 144: “Data analysis materialised a number of themes and subthemes”

Line 494 – change “punish the perpetrators” to “hold perpetrators to account”

Lines 505-507 – re-word “Similar to the current finding……….support services” 

R14. Thank you for highlighting this. 

Revisions were made to remove these highlighted awkward expressions.

C. Reviewer #3: This paper is in an area of great importance because there is a dearth of interventions for IPV especially in low income countries. Process evaluations of trials of interventions are vital and not done enough in research. There are several areas for improvement in the manuscript detailed below.

C1. Abstract

It is not clear what the research questions are or specific objectives. The themes are a bit difficult to interpret 

R1. Thank you for your feedback. The objectives and results in the abstract have been rewritten to make it clearer (Page 2, line no: 25-28; 33-35). 

C2. Introduction is a good background and the objectives are clear here. 

R2. Thank you for the comment

C3. Methods. It is not clear when the second interviews were done in relation to the intervention as women were recruited up to 34 weeks?

How many people in total were in the intervention group- was it 63?

What was the pool of HCPs that the 7 were selected from and how were they able to provide the most pertinent information? 

R3. Thank you for your comment. The second interviews were conducted at 4-6 weeks post-intervention and few women with 34 weeks of gestation were included in this study and they had delivered after 38 weeks of gestation, so having second interview with them was not an issue. 

The number of women recruited was mentioned in the first line of results section (Page 10, line no. 197)

“A total of 63 women at the first follow-up and 51 women at the second follow-up, belonging to the intervention group, were asked about their views and opinions regarding the intervention.”

The characteristics of the HCPs included in the trial were mentioned in the first paragraph of results section (Page 11, line no 200-205).

C4. What were the interview questions? 

R4. Thank you. Interview questions used for participants and health care providers were included in supplementary file 2.

C5. Were the women and HCPs analysed together? and why? 

R5. Yes, women and HCPs were analysed together as most of the themes emerging from the data overlapped and while interpreting the findings, attempts were made to clarify which quotes belonged to which group of participants. For eg: ‘HCP’ was written if that quote was from health care providers and ‘Woman’ was written for the quotes extracted from interview with intervention participants.

C6. Please spell out the Braun and Clarke method in more detail. 

R6. Analysis section has been rewritten based on the 6 steps of thematic analysis proposed by Braun and Clarke (Page no. 9, line no.: 154-172).

C7. Results

It is confusing the domains and then the themes and they are not reflected in the abstract clearly. Further the material needs to be synthesised more there are too many themes. A further synthesis would strengthen the analysis. 

R7. Abstract has mentioned about the broad domains of the qualitative data analysis (Page 2, line no. 33-35). Key findings were presented in the result section.

Figure 1 illustrates domains, themes and subthemes generated from data analysis.

Thank you for your suggestion. Some of the subthemes were synthesized which we believed might have strengthened the analysis now (Page no. 15-20). 

C8. The themes are very descriptive, which does not match quality thematic analyses. It appears that the domains might reflect direct interview questions which has resulted in this very descriptive level analyses? 

R8. Thank you for your suggestion. This study aimed to explore the perceptions and experiences of intervention participants and HCPs regarding the intervention. So, themes reflecting this objective were identified and finalised after iterative discussion with the study authors.

C9. Why is the first domain only from HCPs? 

R9. Intervention participants were asked to provide answer to open-ended questions regarding strengths and weakness of the intervention, and any recommendation they might have during their follow-up assessments. At this point, it was not appropriate to ask about the participants’ views regarding DFV and its response mechanisms. However, in-depth interviews were conducted with the HCPs and question regarding how they perceive the problem of DFV and the existing efforts to address were asked at first and it is deemed important to explore their views regarding DFV before asking them about their views regarding the piloted intervention. 

A sentence has been added in the Methods: Data collection section (Page 8, line no 138-139) and a next sentence has been added in Results: domain 1 to make it clear (Page 11, line 211).

C10. Please remove some of the acronyms 

R10. Acronyms are used as less as possible and are described in full form at the time of their first use.

C11. In the second domain it would be good to get a sense of the strength of the themes- some women, most women.... 

R11. Thank you, we have tried to provide a response where possible (Page 14-16) 

C12. p12 it is not helpful to discuss the quantitative findings as they are not presented here. 

R12. Thank you. The sentence has been restructured as suggested (Page 14, line no. 275).

C13. Some of the quotes are very powerful but sometimes the headings for the subthemes don't match the quotes e.g. new and positive learning experience starts with the value of being asked alone and the feeling empowered section suddenly has barriers using the phone? 

R13. Thank you for your suggestion.

Some subthemes were merged to make them easy to understand (Page no. 15-21).

C14. Discussion

The discussion repeats the findings quite a lot rather than summarising and the conclusion is not helpful as we don’t have the quantitative results. 

R14. Thank you for your comments.

The findings were discussed in brief before providing supporting or contradicting arguments based on other studies. Following your comments, the discussion section has been revised and discussed in more detail with reference to other literature (Page no. 24-27). 

This study summarises the findings of qualitative analysis and conclusion is based on the qualitative findings. Quantitative findings of the study have also been analysed and support the qualitative findings (Manuscript based on quantitative findings is under preparation)

---

## [Decision Letter · Decision Letter 1]

8 Jan 2020

PONE-D-19-22252R1

‘We don’t see because we don’t ask’: qualitative exploration of service users’ and health professionals’ views regarding a psychosocial intervention targeting pregnant women experiencing domestic and family violence

PLOS ONE

Dear Mrs. Sapkota,

Thank you for submitting your manuscript to PLOS ONE. After careful consideration, we feel that it has merit but does not fully meet PLOS ONE’s publication criteria as it currently stands. Therefore, we invite you to submit a revised version of the manuscript that addresses the points raised during the review process.

We would appreciate receiving your revised manuscript by Feb 22 2020 11:59PM. To enhance the reproducibility of your results, we recommend that if applicable you deposit your laboratory protocols in protocols.io, where a protocol can be assigned its own identifier (DOI) such that it can be cited independently in the future. For instructions see: http://journals.plos.org/plosone/s/submission-guidelines#loc-laboratory-protocols

We look forward to receiving your revised manuscript.

Kind regards,

Susan A. Bartels, MD, MPH, FRCPC

Academic Editor

PLOS ONE

Additional Editor Comments (if provided):

Thank you for submitting your revised manuscript. It has been reviewed by two indiviiduals who have made additional suggestions to improve the article, particularly around editing, grammar, choice of language, etc.

Reviewers' comments:

Reviewer's Responses to Questions

**Comments to the Author**

1. If the authors have adequately addressed your comments raised in a previous round of review and you feel that this manuscript is now acceptable for publication, you may indicate that here to bypass the “Comments to the Author” section, enter your conflict of interest statement in the “Confidential to Editor” section, and submit your "Accept" recommendation.

Reviewer #2: (No Response)

Reviewer #4: (No Response)

2. Is the manuscript technically sound, and do the data support the conclusions?

Reviewer #2: Yes

Reviewer #4: Yes

3. Has the statistical analysis been performed appropriately and rigorously? 

Reviewer #2: N/A

Reviewer #4: N/A

4. Have the authors made all data underlying the findings in their manuscript fully available?

Reviewer #2: Yes

Reviewer #4: Yes

5. Is the manuscript presented in an intelligible fashion and written in standard English?

Reviewer #2: Yes

Reviewer #4: No

6. Review Comments to the Author

Reviewer #2: Thank you for addressing issues raised in the previous round of review. Upon re-reviewing the manuscript, however, I would like to raise issues that I feel still require further response.

There appears to be inconsistency in the stated aims of the research.

The abstract (lines 25-26) - "acceptability and the perceived impact of....[the intervention]

The introduction (lines 135-119) - "....perceived strengths and limitations of [the intervention] (described as 'objectives')

Finding - Line 382 - 'improvement and sustainability'. Although this is not an aim per se, it does highlight the inconsistency across the manuscript in what was being addressed through this particular study.

Clarity and consistency needed.

Other inconsistencies:

Line 133, reference is made to "the research nurse", yet line 139 refers to "Both interviewers'. It is unclear how many interviewers there were (1 or 2) and whether the 'research nurse' is one of them. Clarity is needed.

Lines 277 and 282 - the term 'participants' is used. 'Participants' refer to BOTH the HCPs and women. If the discussion is about how the women felt, then I would anticipate that 'participant' should actually be women. A similar issue arises throughout the theme "Impact of intervention on women's lives"

Other issues

Line 169 - what is meant by "reasoned, logical and valid"

Lines 179-180 : it is noted that "A review of field notes in conjunction with the transcripts addressed dependability". Dependability is normally addressed through the keeping of an audit trail. It is unclear how merely reviewing notes and transcripts addresses dependability. Explanation (with references) needed.

Line 274 - 'Verbal feedback' is referred to. Should this not be data from the interviews?

What is meant by 'fair proportion' -line 344

There is still some disconnect between the surrounding text and the quotes provided as evidence:

Eg

Lines 224 - 225. The main thrust of the quote is family relations but the text refers to lack of education

Lines 287 - 288. This quote is about 'confidence'. The link to how a person centred approach enables women to come forward is still unclear.

Lines 289 - 291 - the quote which refers to 'contained space' (I'm assuming is quite removed from where this was discussed in the text. Please consider inserting.

Lines 353 - 354 - Why is there a quote from a woman when the preceding text is related to HCPs?

Lines 429 - 434 - the preceding text refers to collaboration with stakeholders, supportive supervision and debriefing are discussed but different issues are raised in the quotes

Line 463 - how does 'adequate time' related to 'the best quality care' as described in the preceding text.

Written English. Although it is noted that the manuscript has been copy edited, there are a number of instances throughout where further work is required.

Eg:

Abstract:

Lines 36-37 : "Intervention participants expressed the counselling session as a safe haven....." Awkward expression

Line 64 - consider not using the term 'cases' to refer to women' (ie, the first time 'cases' is mentioned in the sentence)

Line 70 - The term 'programmes insufficient' is used. Are the programs insufficient or the number (or spread) of programs insufficient? See also line 83

Line 72 - what is meant by the 'enforcement' of the initiatives. Please consider using a more relevant term

Line 74 - Please reconsider the phrase 'larger segment of population in Nepal' - Awkward expression

Line 79 - I a review 'on' intervention, or 'about' interventions

Line 85 - What is meant by 'alert women against DFV' - Awkward expression.

Line 90 - we screen victims 'ford' DFV not 'of DFV'

Line 96 - change 'involvements' to 'involvement'

Line 151 - "....were read in conjunction with the transcribed verbatim". It appears word/s are missing.

Line 179 - What is meant by "a triangulation of researchers"?

Line 213 - Do you mean,.....'and overlapping in nature'?

Line 256 - change 'currently' to at the time the article was written, as this information may go out of date.

Line 259 - insert 'the' between 'notify' and 'hospital's'

Line 301-302 - ...learn how to respond to FV and some topics

Line 387 0 delete 'as well'

Line 515 - insert 'the' between 'of' and 'booklet'

Line 516- change 'after' to 'rather'

Line 518 - insert 'a' between 'DFV' and 'one-size-fits-all"

Line 520 - change 'influences' to 'influence'

Line 544- meaning of 'awareness on need of acting against DFV' is unclear

Line 545 - delete 'very'

Line 552 - this sentence reads as if the HCPs are disclosing the women's experiences. I'm sure this is not what was intended

Line 559 - delete 'the'

Line 564 - do you mean 'funding in this area'?

Line 573, delete the comma after 'Despite'

Line 581. Start a new sentence at 'this study'

Reviewer #4: This qualitative study examines the acceptability and utility of a psychosocial intervention, from the perspective of service users and health professionals. Overall, the authors have addressed the reviewers’ prior concerns and comments.

I’m not sure why the authors have chosen to use DFV for Domestic & Family Violence since the more common terms (and therefore more searchable by potential readers) are DV (domestic violence) or IPV (intimate partner violence). I recommend the authors use the most commonly used IPV.

While significantly improved over the initial submission, the manuscript requires copyediting for language usage, grammar and clarity. Related to this, some of the quotes were poorly translated into English.

7. PLOS authors have the option to publish the peer review history of their article (what does this mean?). If published, this will include your full peer review and any attached files.

Reviewer #2: No

Reviewer #4: No

---

## [Author Response · Author response to Decision Letter 1]

14 Feb 2020

A. Reviewer 2: Thank you for addressing issues raised in the previous review. Upon re-reviewing the manuscript, however, I would like to raise issues that I feel still require further response.

C1. There appears to be inconsistency in the stated aims of the research.

The abstract (lines 25-26) - "acceptability and the perceived impact of....[the intervention]

Finding - Line 382 - 'improvement and sustainability'. Although this is not an aim per se, it does highlight the inconsistency across the manuscript in what was being addressed through this particular study. Clarity and consistency needed.

The introduction (lines 135-119) - "....perceived strengths and limitations of [the intervention] (described as 'objectives' 

R1: Thank you. We appreciate the reviewer’s comments. 

To assess acceptability of any intervention, it is necessary to identify the perceived impact of that intervention (strengths) and barriers in implementing that intervention. In addition, exploring recommendations and suggestion from participants will help to design the intervention in the most effective way for future implementation. Hence, in the abstract section, general objective was stated and it was further broken down into specific objectives. (Page no. 7, line no. 115-119)

This study has two specific objectives.

• To explore the perceived strengths and limitations of the counselling and psychoeducation intervention as experienced by intervention participants and HCPs.

• To solicit feedback on what elements of the intervention and process could be improved and how.

The objective in the abstract was restructured to make it clearer and consistent. (Page no. 2, line no. 26-28)

As suggested, the sentence was restructured to ensure consistency. (Page no. 19, line no. 388-89)

C2. Line 133, reference is made to "the research nurse", yet line 139 refers to "Both interviewers'. It is unclear how many interviewers there were (1 or 2) and whether the 'research nurse' is one of them. Clarity is needed. 

R2: Thank you for the information.

The research nurse (RP) had interviewed participants from the intervention group and the intervention nurse (DS) had interviewed HCPs. 

This had been mentioned in the Data Collection section and to make it clearer, initials of both interviewers have been added in the text.

(Page no. 8, line no. 134, 137, 141)

C3. Lines 277 and 282 - the term 'participants' is used. 'Participants' refer to BOTH the HCPs and women. If the discussion is about how the women felt, then I would anticipate that 'participant' should actually be women. A similar issue arises throughout the theme "Impact of intervention on women's lives" 

R3: Thank you for your feedback.

This study had explored perspectives of both the service providers and service users. It is not just about how women felt but also how service providers felt about the intervention. Hence, ‘participants’ term was used to indicate both of them and either woman or HCP was used after each quote to make the reader better understand which quote was said by whom.

C4. Line 169 - what is meant by "reasoned, logical and valid" 

R4: Thank you for pointing this out.

The interpretation made in the result section was rigorously discussed among the research team to ensure it is logical and valid. Furthermore, appropriate reasoning was put forward to support the findings presented.

As suggested the sentence was restructured to make it clearer. (Page no. 9; line no. 171)

C5. Lines 179-180 : it is noted that "A review of field notes in conjunction with the transcripts addressed dependability". Dependability is normally addressed through the keeping of an audit trail. It is unclear how merely reviewing notes and transcripts addresses dependability. Explanation (with references) needed. 

R5: Thank you for pointing this out. 

This section has been amended. (Page no. 10; line no. 182-84)

C6. Line 274 - 'Verbal feedback' is referred to. Should this not be data from the interviews? 

What is meant by 'fair proportion' -line 344 

R6: Thank you for your feedback. This has been revised as suggested. (Page no. 14; line no. 279). The proportion is now quantified to make it clearer. (Page no. 17; line no. 348)

C7. There is still some disconnect between the surrounding text and the quotes provided as evidence:

Eg

Lines 224 - 225. The main thrust of the quote is family relations but the text refers to lack of education 

R7: Thank you for your feedback. In this section, some contributing factors of DFV as expressed by participants were presented. The sentence was restructured to make it clearer. (Page no. 12; line no. 226-227)

Other quotes were also reviewed and arranged appropriately to ensure that the surrounding text align with the quotes provided.

C8. Lines 287 - 288. This quote is about 'confidence'. The link to how a person centred approach enables women to come forward is still unclear. 

R8: Thank you for your feedback.

Person-centered approach would encourage women to express themselves without hesitation and address their individual needs and concerns. This approach would improve confidence among women to share their problem with HCPs.

C9. Lines 289 - 291 - the quote which refers to 'contained space' (I'm assuming is quite removed from where this was discussed in the text. Please consider inserting. 

R9: Thank you for your feedback. There is a quote referring to contained space in Page no. 15; line no. 294-296.

C10. Lines 353 - 354 - Why is there a quote from a woman when the preceding text is related to HCPs? 

R10: Thank you for your feedback. Explanatory text was added before the quote to make it clearer. (Page no. 18; line no. 358-59)

C11. Lines 429 - 434 - the preceding text refers to collaboration with stakeholders, supportive supervision and debriefing are discussed but different issues are raised in the quotes 

R11: Thank you for your feedback. 

‘All higher authorities need to be involved’ (Page no. 21; line no. 440) – This implies collaboration with the stakeholders at all level for proper implementation of the program.

C12. Line 463 - how does 'adequate time' related to 'the best quality care' as described in the preceding text

R12: When there will be a decrease in patient load, health care providers will get adequate time for discussing with patients, which is crucial for improving the effective doctor-patient relationship. This approach would ensure correct identification and management of the patient’s issue. However, failure to allocate adequate time for patient’s consultation might raise several issues such as misdiagnosis, miscommunication and lack of empathy from the doctor leading to distrust from the patients. 

(Ref: https://praxhub.com/are-your-patients-receiving-adequate-time-and-quality-care/)

C13. Written English. Although it is noted that the manuscript has been copy edited, there are a number of instances throughout where further work is required. 

 Eg:

Abstract: 

Lines 36-37 : "Intervention participants expressed the counselling session as a safe haven....." Awkward expression 

Thank you for your feedback.

The sentence has been restructured to make it clearer. (Page no. 2; line no. 37-38)

Line 64 - consider not using the term 'cases' to refer to women' (ie, the first time 'cases' is mentioned in the sentence) 

Thank you for your feedback. The suggested change has been made. (Page no. 4; line no. 62)

Line 70 - The term 'programmes insufficient' is used. Are the programs insufficient or the number (or spread) of programs insufficient? See also line 83 

Thank you for your feedback. The sentence has been restructured to make it clearer. (Page no. 4; line no. 70-72; Page no. 5; line no 83-84)

Line 72 - what is meant by the 'enforcement' of the initiatives. Please consider using a more relevant term 

Thank you for your feedback. The sentence has been restructured to make it clearer. (Page no. 4; line no. 72-73)

Line 74 - Please reconsider the phrase 'larger segment of population in Nepal' - Awkward expression 

Thank you for your feedback. 

The sentence has been restructured to make it clearer. (Page no. 5; line no. 74-75)

Line 79 - I a review 'on' intervention, or 'about' interventions 

Thank you. Change has been made as suggested. (Page no. 5; line no. 79)

Line 85 - What is meant by 'alert women against DFV' - Awkward expression. 

Thank you for your feedback. The sentence has been restructured to make it clearer. (Page no. 5; line no. 85)

Line 90 - we screen victims 'ford' DFV not 'of DFV' 

Thank you. Change has been made as suggested. (Page no. 5; line no. 90)

Line 96 - change 'involvements' to 'involvement' 

Thank you. Change has been made as suggested. (Page no. 5; line no. 96)

Line 151 - "....were read in conjunction with the transcribed verbatim". It appears word/s are missing. 

Thank you for your feedback. The sentence has been restructured to make it clearer. (Page no. 8-9; line no. 152-54)

Line 179 - What is meant by "a triangulation of researchers"? 

Triangulation of researchers means involvement of multiple researchers/observers with different areas of expertise. This approach enables securing as many different views as possible during the analysis of the qualitative data (Denzin, 1978). Available in https://qualpage.com/2018/01/18/triangulation-in-qualitative-research/

Line 213 - Do you mean,.....'and overlapping in nature'? 

Thank you. It has now been corrected. (Page no. 11; line no. 217)

Line 256 - change 'currently' to at the time the article was written, as this information may go out of date. 

Thank you. It has now been corrected. (Page no. 13; line no. 258-59)

Line 259 - insert 'the' between 'notify' and 'hospital's' 

Thank you. Change has been made as suggested. (Page no. 13; line no. 263)

Line 301-302 - ...learn how to respond to FV and some topics 

Thank you. We appreciate the reviewer’s comments. We have rechecked the line highlighted to ensure it is correct.

Line 387 0 delete 'as well' 

Thank you. As suggested, ‘as well’ was deleted. (Page no. 19; line no. 394)

Line 515 - insert 'the' between 'of' and 'booklet' 

Thank you. As suggested, ‘the’ was added. (Page no. 25; line no. 522)

Line 516- change 'after' to 'rather' 

Thank you. Change has been made as suggested. (Page no. 25; line no. 524)

Line 518 - insert 'a' between 'DFV' and 'one-size-fits-all" 

Thank you. Change has been made as suggested. (Page no. 25; line no. 526)

Line 520 - change 'influences' to 'influence' 

Thank you. Change has been made as suggested. (Page no. 25; line no. 528)

Line 544- meaning of 'awareness on need of acting against DFV' is unclear 

Thank you for your feedback. The sentence has been restructured to make it clearer. (Page no. 26; line no. 552-54)

Line 545 - delete 'very' 

Thank you. The necessary amendment was made. (Page no. 26; line no. 553)

Line 552 - this sentence reads as if the HCPs are disclosing the women's experiences. I'm sure this is not what was intended Thank you. The sentence has now been restructured to make it clearer. (Page no. 27; line no. 559-62)

Line 559 - delete 'the' 

Thank you. The necessary amendment was made. (Page no. 27; line no. 568)

Line 564 - do you mean 'funding in this area'? 

Thank you. The necessary amendment was made. (Page no. 27; line no. 573)

Line 573, delete the comma after 'Despite' 

Thank you. As suggested, a comma was added. (Page no. 28; line no. 582)

Line 581. Start a new sentence at 'this study' 

Thank you for your feedback (Page no. 28; line no. 594)

B. Reviewer’s 4 comments 

 This qualitative study examines the acceptability and utility of a psychosocial intervention, from the perspective of service users and health professionals. Overall, the authors have addressed the reviewers’ prior concerns and comments.

C1. I’m not sure why the authors have chosen to use DFV for Domestic & Family Violence since the more common terms (and therefore more searchable by potential readers) are DV (domestic violence) or IPV (intimate partner violence). I recommend the authors use the most commonly used IPV. 

R1: Thank you. We appreciate the reviewer’s suggestion. We do agree that IPV is the most common term, however, all of the authors have had discussions about the best terminology to be used in the article at the time of the protocol development. Agreement on using DFV was based on the understanding that in the setting where the study was conducted, (i.e. Nepal, the perpetrators include not only the husband, but other family members, including father, mother, and brother-in-laws. This has been described in the protocol paper which has been published elsewhere. The link to the protocol paper is

https://bmjopen.bmj.com/content/9/4/e027436

C2. While significantly improved over the initial submission, the manuscript requires copyediting for language usage, grammar and clarity. Related to this, some of the quotes were poorly translated into English. 

R2: The manuscript has been thoroughly reviewed and grammatical errors have been rectified. Two of the authors are native English speakers who have reviewed again for English edits.

---

## [Editor Report · Decision Letter 2]

21 Feb 2020

‘We don’t see because we don’t ask’: qualitative exploration of service users’ and health professionals’ views regarding a psychosocial intervention targeting pregnant women experiencing domestic and family violence

PONE-D-19-22252R2

Dear Dr. Sapkota,

We are pleased to inform you that your manuscript has been judged scientifically suitable for publication and will be formally accepted for publication once it complies with all outstanding technical requirements.

With kind regards,

Susan A. Bartels, MD, MPH, FRCPC

Academic Editor

PLOS ONE
---

## [Editor Report · Acceptance letter]

26 Feb 2020

PONE-D-19-22252R2 

‘<I>We don’t see because we don’t ask</I>’: qualitative exploration of service users’ and health professionals’ views regarding a psychosocial intervention targeting pregnant women experiencing domestic and family violence 

Dear Dr. Sapkota:

I am pleased to inform you that your manuscript has been deemed suitable for publication in PLOS ONE. Congratulations! Your manuscript is now with our production department. 

With kind regards,

on behalf of

Dr. Susan A. Bartels 

Academic Editor

PLOS ONE